# Fictitious Play for Mean Field Games:
# Continuous Time Analysis and Applications

**Sarah Perrin**[*,1], **Julien Perolat**[*,2], **Mathieu Laurière**[3], **Matthieu Geist**[4],

**Romuald Elie**[2], **Olivier Pietquin**[4]

Univ. Lille, CNRS, Inria, UMR 9189 CRIStAL[1]        DeepMind Paris[2]

Princeton University, ORFE[3]        Google Research, Brain Team[4]

sarah.perrin@inria.fr        perolat@google.com

lauriere@princeton.edu        [mfgeist, relie, pietquin]@google.com

## Abstract

In this paper, we deepen the analysis of continuous time Fictitious Play learning algorithm to the consideration of various finite state Mean Field Game settings (finite horizon, $\gamma$-discounted), allowing in particular for the introduction of an additional common noise. We first present a theoretical convergence analysis of the continuous time Fictitious Play process and prove that the induced exploitability decreases at a rate $O(\frac{1}{t})$. Such analysis emphasizes the use of exploitability as a relevant metric for evaluating the convergence towards a Nash equilibrium in the context of Mean Field Games. These theoretical contributions are supported by numerical experiments provided in either model-based or model-free settings. We provide hereby for the first time converging learning dynamics for Mean Field Games in the presence of common noise.

## 1   Introduction

Learning in games has a long history [103, 101] but learning in the midst of a large number of players still remains intractable. Even the most recent successes of machine learning, including Reinforcement Learning (RL) [112], remain limited to interactions with a handful of players (*e.g.* Go [106, 108, 107], Chess [28], Checkers [102, 101], Hex [13], Starcraft II [114], poker games [24, 25, 91, 21] or Stratego [87]). Whilst the general multi-agent learning case might seem out of reach, considering interactions within a very large population of players may lead to tractable models. Inspired by the large economic literature on games with a continuum of players [15], the notion of Mean Field Games (MFGs) has been introduced in [84, 76] to model strategic interactions through the distribution of players' states. In such framework, all players are identical, anonymous (*i.e.*, they are not identifiable) and have symmetric interests. In this asymptotic formulation, the learning problem can be reduced to characterizing the optimal interactions between one representative player and the full population.

Most of the MFG literature assumes the representative player to be fully informed about the game dynamics and the associated reward mechanisms. In such context, the Nash equilibrium for an MFG is usually computed via the solution of a coupled system of dynamical equations. The first equation models the forward dynamics of the population distribution, while the second is the dynamic programming equation of the representative player. Such approaches typically rely on partial differential equations and require deterministic numerical approximations [9] (*e.g.*, finite differences

---

[*]Equal contribution

methods [4, 3], semi-Lagrangian schemes [34, 35], or primal-dual methods [23, 22]). Despite the success of these schemes, an important pitfall for applications is their lack of scalability. In order to tackle this limitation, stochastic methods based on approximations by neural network have recently been introduced in [39, 40, 59] using optimality conditions for general mean field games, in [98] for MFGs which can be written as a control problem, and in [29, 86] for variational MFGs in connection with generative adversarial networks. We now contribute and take a new step forward in this direction.

We investigate a generic and scalable simulation-based learning algorithm for the computation of approximate Nash equilibria, building upon the Fictitious Play scheme [97, 60, 104]. We study the convergence of Fictitious Play for MFGs, using tools from the continuous learning time analysis [71, 93, 73]. We derive a convergence of the Fictitious Play process at a rate $O(\frac{1}{t})$ in finite horizon or over $\gamma$-discounted monotone MFGs (see Appx. E), thus extending previous convergence results restricted to simpler games [71]. Besides, our approach covers games where the players share a common source of risk, which are widely studied in the MFG literature and crucial for applications. To the best of our knowledge, we derive for the first time convergence properties of a learning algorithm for these so-called MFGs with common noise (where a common source of randomness affects all players [36]). Furthermore, our analysis emphasizes the role of *exploitability* as a relevant metric for characterizing the convergence towards a Nash equilibrium, whereas most approximation schemes in the MFG literature quantify the rate of convergence of the population empirical distribution. The contribution of this paper is thus threefold: (1) we provide several theoretical results concerning the convergence of *continuous time* Fictitious Play in MFGs matching the $O(\frac{1}{t})$ rate existing in zero-sum two-player normal form game, (2) we generalize the notion of *exploitability* to MFGs and we show that it is a meaningful metric to evaluate the quality of a learned control in MFGs, and (3) we empirically illustrate the performance of the resulting algorithm on several MFG settings, including examples with *common noise*.

## 2   Background on Finite Horizon Mean Field Games

A Mean Field Game (MFG) is a temporally extended decision making problem involving an infinite number of identical and anonymous players. It can be solved by focusing on the optimal policy of a representative player in response to the behavior of the entire population. Let $\mathcal{X}$ and $\mathcal{A}$ be finite sets representing respectively the state and action spaces. The representative player starts the game in state $x \in \mathcal{X}$ according to an initial distribution $\mu_0$ over $\mathcal{X}$. At each time step $n \in [0, \ldots, N]$, the representative player being in state $x_n$ takes an action $a_n$ according to a policy $\pi_n(a_n|x_n)$. As a result, the player moves to state $x_{n+1}$ according to the transition probability $p(.|x_n, a_n)$ and receives a reward $r(x_n, a_n, \mu_n)$, where $\mu_n$ represents the distribution over states of the entire population at time $n$. For a given sequence of policies $\pi = (\pi_n)_n$ and a given sequence of distributions $\mu = (\mu_n)_n$, the representative player will receive the cumulative sum of rewards defined as[2]:

$$J(\mu_0, \pi, \mu) = \mathbb{E}\left[\sum_{n=0}^{N} r(x_n, a_n, \mu_n) \mid x_0 \sim \mu_0, \ x_{n+1} = p(.|x_n, a_n), \ a_n \sim \pi_n(.|x_n)\right].$$

$Q$-**functions and value functions:** The $Q$-function is defined as the expected sum of rewards starting from state $x$ and doing action $a$ at time $n$:

$$Q_n^{\pi, \mu}(x, a) = \mathbb{E}\left[\sum_{k=n}^{N} r(x_k, a_k, \mu_k) \mid x_n = x, \ a_n = a, \ x_{k+1} = p(.|x_k, a_k), \ a_k \sim \pi_k(.|x_k)\right].$$

By construction, it satisfies the recursive equation:

$$Q_N^{\pi, \mu}(x, a) = r(x, a, \mu_N), \quad Q_{n-1}^{\pi, \mu}(x, a) = r(x, a, \mu_{n-1}) + \sum_{x' \in \mathcal{X}} p(x'|x, a)\mathbb{E}_{b \sim \pi_n(.|x')}\left[Q_n^{\pi, \mu}(x', b)\right].$$

The value function is the expected sum of rewards for the player that starts from state $x$ and can thus be defined as: $V_n^{\pi, \mu}(x) = \mathbb{E}_{a \sim \pi(.|x)}\left[Q_n^{\pi, \mu}(x, a)\right]$. Note that the objective function $J$ of a representative player rewrites in particular as an average at time 0 of the value function $V$ under the initial distribution $\mu_0$: $J(\mu_0, \pi, \mu) = \mathbb{E}_{x \sim \mu_0(.)}\left[V_0^{\pi, \mu}(x)\right]$.

**Distribution induced by a policy:** The state distribution induced by $\pi = \{\pi_n\}_n$ is defined recursively by the forward equation starting from $\mu_0^\pi(x) = \mu_0(x)$ and $\mu_{n+1}^\pi(x') = \sum_{x,a \in \mathcal{X} \times \mathcal{A}} \pi_n(a|x)p(x'|x,a)\mu_n^\pi(x)$.

**Best Response:** A best response policy $\pi^{BR}$ is a policy that satisfies $J(\mu_0, \pi^{BR}, \mu^\pi) = \max_{\pi'} J(\mu_0, \pi', \mu^\pi)$. Intuitively, it is the optimal policy an agent could take if it was to deviate from the crowd's policy.

**Exploitability:** The exploitability $\phi(\pi)$ of policy $\pi$ quantifies the average gain for a representative player to replace its policy by a best response, while the entire population plays with policy $\pi$: $\phi(\pi) := \max_{\pi'} J(\mu_0, \pi', \mu^\pi) - J(\mu_0, \pi, \mu^\pi)$. Note that, as it scales with rewards, the absolute value of the exploitability is not meaningful. What matters is its relative value compared with a reference point, such as the exploitability of the policy at initialization of the algorithm. In fact, the exploitability is game dependent and hard to re-scale without introducing other issues (dependence on the initial policy if we re-normalize with the initial exploitability for example).

**Nash equilibrium:** A Nash equilibrium is a policy satisfying $\phi(\pi) = 0$ while an approximate Nash equilibrium has a small level of exploitability.

The exploitability is an already well known metrics within the computational game theory literature [117, 21, 83, 26], and one of the objectives of this paper is to emphasize its important role in the context of MFGs. Classical ways of evaluating the performance of numerical methods in the MFG literature typically relate to distances between distribution $\mu$ or value function $V$, as for example in [9]. A close version of the exploitability has been used in this context (*e.g.*, [68]), but being computed over all possible starting states at any time. Such formulation gives too much importance to each state, in particular those having a (possibly very) small probability of appearance. In comparison, the exploitability provides a well balanced average metrics over the trajectories of the state process.

**Monotone games:** A game is said monotone if the reward has the following structure: $r(x,a,\mu) = \tilde{r}(x,a) + \bar{r}(x,\mu)$ and $\forall \mu, \mu'$, $\sum_{x \in \mathcal{X}}(\mu(x) - \mu'(x))(\bar{r}(x,\mu) - \bar{r}(x,\mu')) \leq 0$. This so-called Lasry-Lions monotonicity condition is classical to ensure the uniqueness of the Nash equilibrium [84].

**Learning in finite horizon problems:** When the distribution $\mu$ of the population is given, the representative player faces a classical finite horizon Markov Decision problem. Several approaches can be used to solve this control problem such as model-based algorithms (*e.g.* backward induction: Algorithm 4 in Appx. D, with update rule $\forall a, x \in \mathcal{A} \times \mathcal{X}$ $Q_{n-1}^\mu(x,a) = r(x,a,\mu_{n-1}) + \sum_{x' \in \mathcal{X}} p(x'|x,a) \max_b Q_n^\mu(x',b))$ or model-free algorithms (*e.g.* $Q$-learning: Algorithm 2 in Appx. D with update rule $Q_n^{k+1}(x_n^k, a_n^k) = (1-\alpha)Q_n^{k+1}(x_n^k, a_n^k) + \alpha[r(x_n^k, a_n^k, \mu_{k-1}) + \max_b Q_{n+1}^k(x_{n+1}^k, b)])$.

**Computing the population distribution:** Once a candidate policy is identified, one needs to be able to compute (or estimate) the induced distribution of the population at each time step. It can either be computed exactly using a model-based method such as Algorithm 5 in Appx. D, or alternatively be estimated with a model-free method like Algorithm 3 in Appx. D.

**Fictitious Play for MFGs:** Consider available (1) a computation scheme for the population distribution given a policy, and (2) an approximation algorithm for an optimal policy of the representative player in response to a population distribution. Then, discrete time Fictitious Play presented in Algorithm 1 provides a robust approximation scheme for Nash equilibrium by computing iteratively the best response against the distribution induced by the average of the past best responses. We will analyse this discrete time process in continuous time in section 3. To differentiate the discrete time from the continuous time, we denote the discrete time with $j$ and the continuous time with $t$. At a given step $j$ of Fictitious Play, we have that:

$$\forall n, \ \bar{\mu}_n^j = \frac{j-1}{j}\bar{\mu}_n^{j-1} + \frac{1}{j}\mu_n^{\pi^j}$$

The policy generating this average distribution is:

$$\forall n, \ \bar{\pi}_n^j(a|x) = \frac{\sum_{i=0}^j \mu_n^{\pi^i}(x)\pi_n^i(a|x)}{\sum_{i=0}^j \mu_n^{\pi^i}(x)}.$$

**Algorithm 1:** Fictitious Play in Mean Field Games

---

**input** : Start with an initial policy $\pi_0$, an initial distribution $\mu_0$ and define $\bar{\pi}_0 = \pi_0$

**1 for** $j = 1, \ldots, J$: **do**

**2**      find $\pi^j$ a best response against $\bar{\mu}^j$ (either with $Q$-learning or with backward induction);

**3**      compute $\bar{\pi}^j$ the average of $(\pi^0, \ldots, \pi^j)$;

**4**      compute $\mu^{\pi^j}$ (either with a model-free or model-based method);

**5**      compute $\bar{\mu}^j$ the average of $(\mu^0, \ldots, \mu^{\pi^j})$

**6 return** $\bar{\pi}^J$, $\bar{\mu}^J$

---

## 3 Continuous Time Fictitious Play in Mean Field Games

In this section, we study a continuous time version of Algorithm 1. The continuous time Fictitious Play process is defined following the lines of [71, 93]. First, we start for $t < 1$ with a fixed policy $\bar{\pi}^{t<1} = \{\bar{\pi}_n^{t<1}\}_n = \{\pi_n^{t<1}\}_n$ with induced distribution $\bar{\mu}^{t<1} = \mu^{t<1} = \mu^{\pi^{t<1}} = \{\mu_n^{\pi^{t<1}}\}_n$ (this arbitrary policy for $t \in [0, 1]$ is necessary for the process to be defined at the starting point). Then, the Fictitious Play process is defined for all $t \geq 1$ and $n \in [1, \ldots, N]$ as:

$$\frac{d}{dt}\bar{\mu}_n^t(x) = \frac{1}{t}\left(\mu_n^{\mathrm{BR},t}(x) - \bar{\mu}_n^t(x)\right) \quad \text{or in integral form:} \quad \bar{\mu}_n^t(x) = \frac{1}{t}\int_{s=0}^{t} \mu_n^{\mathrm{BR},s}(x)ds\,,$$

where $\mu_n^{\mathrm{BR},t}$ denotes the distribution induced by a best response policy $\{\pi_n^{\mathrm{BR},t}\}_n$ against $\bar{\mu}_n^t(x)$. Hence, the distribution $\mu_n^t(x)$ identifies to the population distribution induced by the averaged policy $\{\pi_n^t\}_n$ defined as follows (proof in A):

$$\forall n, \ \bar{\mu}_n^t(x)\frac{d}{dt}\bar{\pi}_n^t(a|x) = \frac{1}{t}\mu_n^{\mathrm{BR},t}(x)[\pi_n^{\mathrm{BR},t}(a|x) - \bar{\pi}_n^t(a|x)]$$

$$\text{or in integral form: } \forall n, \ \bar{\pi}_n^t(a|x)\int_{s=0}^{t}\mu_n^{\mathrm{BR},s}(x)ds = \int_{s=0}^{t}\mu_n^{\mathrm{BR},s}(x)\pi_n^{\mathrm{BR},s}(a|x)ds,$$

with $\pi_n^{\mathrm{BR},s}$ being chosen arbitrarily for $t \leq 1$. We are now in position to provide the main result of the paper quantifying the convergence rate of the continuous Fictitious Play process.

**Theorem 1.** *If the MFG satisfies the monotony assumption, we can show that the exploitability is a strong Lyapunov function of the system,* $\forall t \geq 1$: $\frac{d}{dt}\phi(\bar{\pi}^t) \leq -\frac{1}{t}\phi(\bar{\pi}^t)$. *Hence* $\phi(\bar{\pi}^t) = O(\frac{1}{t})$.

The proof of the theorem is postponed to Appendix A. Furthermore, a similar property for $\gamma$ discounted MFGs is provided in Appendix C. We chose to present an analysis in continuous time because it provides convenient mathematical tools allowing to exhibit state of the art convergence rate. In discrete time, similarly to normal form games [78, 45], we conjecture that the convergence rate for monotone MFGs is $O(t^{-\frac{1}{2}})$.

## 4 Experiments on Fictitious Play in the Finite Horizon Case

In this section, we illustrate the theoretical convergence of continuous time Fictitious Play by looking at the discrete time implementation of the process. We focus on classical linear quadratic games which have been extensively studied [20, 64, 49] and for which a closed form solution is available. We then turn to a more difficult numerical setting for experiments[3]. We chose either a full model-based implementation or a full model-free approach of Alg. 1. The model-based uses Backward Induction (Alg. 4) and an exact calculation of the population distribution (Alg. 3). The model-free approach uses $Q$-learning (Alg. 2) and a sampling-based estimate of the distribution (Alg. 5).

## 4.1 Linear Quadratic Mean Field Game

**Environment:** We consider a Markov Decision Process a finite action space $\mathcal{A} = \{-M, \ldots, M\}$ together with a one dimensional finite state space domain $\mathcal{X} = \{-L, \ldots, L\}$, which can be viewed as a truncated and discretized version of $\mathbb{R}$. The dynamics of a typical player picking action $a_n$ at time $n$ are governed by the following equation:

$$x_{n+1} = x_n + (K(m_n - x_n) + a_n)\Delta_n + \sigma\epsilon_n\sqrt{\Delta_n} ,$$

allowing the representative player to either stay still or move to the left or to the right. In order to make the model more complex, an additional discrete noise $\epsilon_n$ can also push the player to the left or to the right with a small probability: $\epsilon_n \sim \mathcal{N}(0,1)$, which is in practice discretized over $\{-3\sigma, \ldots, 3\sigma\}$. The resulting state $x_{n+1}$ is rounded to the closest discrete state.

At each time step, the player can move up to $M$ nodes and it receives the reward:

$$r(x_n, a_n, \mu_n) = [-\frac{1}{2}|a_n|^2 + qa_n(m_n - x_n) - \frac{\kappa}{2}(m_n - x_n)^2]\Delta_n$$

where $m_n = \sum_{x \in \mathcal{X}} x\mu_n(x)$ is the first moment of the state distribution $\mu_n$. $\Delta_n$ is the time lapse between two successive steps, while $q$ and $\kappa$ are given non-negative constants. The first term quantifies the action cost, while the two last ones encourage the player to remain close to the average state of the population at any time. Hereby, the optimal policy pushes each player in the direction of the population average state. We set the terminal reward to $r(x_N, a_N, \mu_N) = -\frac{c_{\text{term}}}{2}(m_N - x_N)^2$.

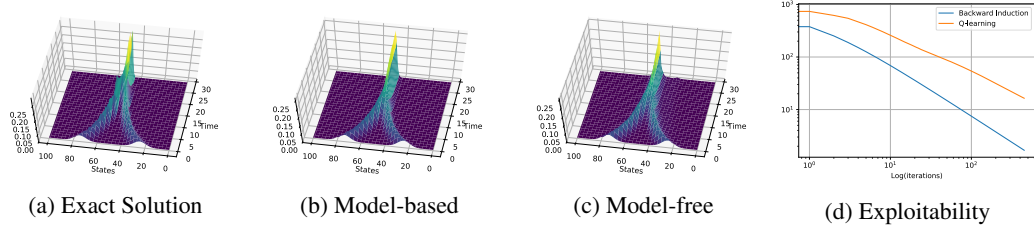

(a) Exact Solution      (b) Model-based      (c) Model-free      (d) Exploitability

Figure 1: Evolution of the distribution in the linear quadratic MFG with finite horizon.

**Experimental setup:** We consider a Linear Quadratic MFG with 100 states and an horizon $N = 30$, which provides a closed-form solution for the continuous state and action version of the game (see Appx. C) and bounds the number of actions $M = 37$ required in the implementation. In practice, the variance $\sigma$ of the idiosyncratic noise $\epsilon_n$ is adapted to the number of states. Here, we set $\sigma = 3$, $\Delta_n = 0.1$, $K = 1$, $q = 0.01$, $\kappa = 0.5$ and $c_{\text{term}} = 1$. In all the experiments, we set the learning rate $\alpha$ of $Q$-learning to $0.1$ and the $\varepsilon$-greedy exploration parameter to $0.2$.

**Numerical results:** Figure 1 illustrates the convergence of Fictitious Play model-based and model-free algorithm in such context. The initial distribution, which is set to two separated bell-shaped distributions, are both driven towards $m$ and converge to a unique bell-shaped distribution as expected. The parameter $\sigma$ of the idiosyncratic noise influences the variance of the final normal distribution. We can observe that both Backward Induction and $Q$-learning provide policies that approximate this behaviour, and that the exploitability decreases with a rate close to $O(1/t)$ in the case of the model-based approach, while the model-free decreases more slowly.

## 4.2 The Beach Bar Process

As a second illustration, we now consider the beach bar process, a more involved monotone second order MFG with discrete state and action spaces, that does not offer a closed-form solution but can be analyzed intuitively. This example is a simplified version of the well known Santa Fe bar problem, which has received a strong interest in the MARL community, see e.g. [14, 55].

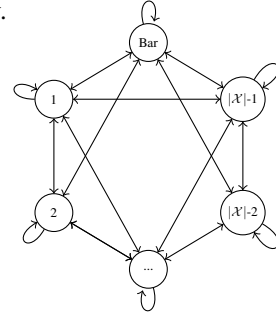

Figure 2: The beach bar process.

**Environment:** The beach bar process (Figure 2) is a Markov Decision Process with $|\mathcal{X}|$ states disposed on a one dimensional

torus ($\mathcal{X} = \{0, \ldots, |\mathcal{X}| - 1\}$), which represents a beach. A bar is located in one of the states. As the weather is very hot, players want to be as close as possible to the bar, while keeping away from too crowded areas. Their dynamics is governed by the following equation:

$$x_{n+1} = x_n + b(x_n, a_n) + \epsilon_n$$

where $b$ is the drift, allowing the representative player to either stay still or move one node to the left or to the right. The additional noise $\epsilon_n$ can push the player one node away to the left or to the right with a small probability:

$$b(x_n, a_n) = \begin{cases} 1 & \text{if } a_n = \text{right} \\ 0 & \text{if } a_n = \text{still} \\ -1 & \text{if } a_n = \text{left} \end{cases} \qquad \epsilon_n = \begin{cases} 1 & \text{with probability } \frac{1-p}{2} \\ 0 & \text{with probability } p \\ -1 & \text{with probability } \frac{1-p}{2} \end{cases}$$

Therefore, the player can go up to two nodes right or left and it receives, at each time step, the reward:

$$r(x_n, a_n, \mu_n) = \tilde{r}(x_n) - \frac{|a_n|}{|\mathcal{X}|} - \log(\mu_n(x_n)) \, ,$$

where $\tilde{r}(x_n)$ denotes the distance to the bar, whereas the last term represents the aversion of the player for crowded areas in the spirit of [11].

**Numerical results:** We conduct an experiment with 100 states and an horizon $N = 15$. Starting from a uniform distribution, we can observe in Figure 3 that both backward induction and $Q$-learning algorithms converge quickly to a peaky distribution where the representative player intends to be as close as possible to the bar while moving away if the bar is already too crowded. The exploitability offers a nice way to measure how close we are from the Nash equilibrium and shows as expected that the model-based algorithm (backward induction) converges at a rate $O(1/t)$ and faster than the model-free algorithm ($Q$-learning).

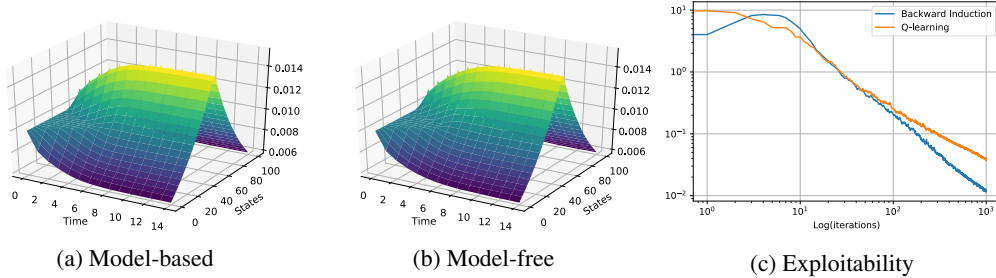

(a) Model-based       (b) Model-free       (c) Exploitability

Figure 3: Beach bar process in finite horizon: (a, b) evolution of the distribution, (c) exploitability.

## 5 Finite Horizon Mean Field Games with Common Noise

We now turn to the consideration of so-called MFG with common noise, that is including an additional discrete *and common* source of randomness in the dynamics. Players still sequentially take actions ($a \in \mathcal{A}$) in a state space $\mathcal{X}$, but the dynamics and the reward are affected by a common noise sequence $\{\xi_n\}_{0 \le n \le N}$. We denote $\Xi_n = \{\xi_k\}_{0 \le k < n} = \Xi_{n-1}.\xi_{n-1}$ where $|\Xi_n|$ represents the total length of the sequence. The extra common source of randomness $\xi$ affects both the reward $r(x, a, \mu, \xi)$ and the probability transition function $p(x'|x, a, \xi)$. We consider policies $\pi_n(a|x, \Xi)$ and population distribution $\mu_n(x|\Xi)$ which are both noise-dependent, and will simply be denoted $\pi_{n,\Xi}(a|x)$ and $\mu_{n|\Xi}(x)$. The $Q$ function is defined as:

$$Q_N^{\pi,\mu}(x, a|\Xi_N) = r(x, a, \mu_{N|\Xi_N}, \xi_N), \quad Q_{n-1}^{\pi,\mu}(x, a|\Xi_{n-1}) = \sum_\xi P(\xi_{n-1} = \xi|\Xi_{n-1}) \Big[$$

$$r(x, a, \mu_{n-1,\Xi_{n-1}}, \xi) + \sum_{x' \in \mathcal{X}} p(x'|x, a, \xi) \mathbb{E}_{b \sim \pi_n(.|x', \Xi_{n-1}.\xi)} \left[ Q_n^{\pi,\mu}(x', b|\Xi_{n-1}.\xi) \right] \Big],$$

while the value function is simply $V_n^{\pi,\mu}(x, \Xi_n) = \mathbb{E}_{a \sim \pi_{n,\Xi_n}(.|x)} \left[ Q_n^{\pi,\mu}(x, a|\Xi_n) \right]$. Similarly, the distribution over states is conditioned on the sequence of noises and satisfies the balance

equation: $\mu_0^\pi(x, \Xi_0) = \mu_0(x)$ (with $\Xi_0$ being the empty sequence $\{\}$) and $\mu_{n+1}^\pi(x'|\Xi.\xi) = \sum_{x\in\mathcal{X}} p^{\pi_{n},\Xi.\xi}(x'|x,\xi)\mu_n^\pi(x|\Xi)$. The expected return for a representative player starting at $\mu_0$ is:

$$J(\mu_0, \pi, \mu) = \sum_{x\in\mathcal{X}} \mu_0(x)V_0^{\pi,\mu}(x, \Xi_0) = \sum_{n=0}^{N} \sum_{\Xi,\xi,|\Xi|=n} P(\Xi.\xi) \sum_{x\in\mathcal{X}} [\mu_n(x, \Xi)r(x, a, \mu_{n,\Xi}, \xi)]$$

with $P(\Xi_0) = 1$ and $P(\Xi.\xi) = P(\xi|\Xi)P(\Xi)$. Finally the **exploitability** is again defined as:

$$\phi(\pi) = \max_{\pi'} J(\mu_0, \pi', \mu^\pi) - J(\mu_0, \pi, \mu^\pi).$$

**Continuous time Fictitious Play for MFGs with common noise:** The Fictitious play process on MFGs with common noise is as follows. For $t < 1$, we start with an arbitrary policy $\bar{\pi}^{t<1}$ (by convention we will take $\bar{\pi}^t = \pi^{\text{BR},t}$ for $t < 1$) whose distribution is $\bar{\mu}^{t<1} = \mu^{\pi^{t<1}}$ (with the convention that $\bar{\mu}^t = \mu^{\text{BR},t}$). Then, for all $t$ and $\Xi$:

$$\bar{\mu}_n^t(x|\Xi) = \frac{1}{t} \int_{s=0}^{t} \mu_n^{\text{BR},s}(x|\Xi)ds,$$

where $\mu^{\text{BR},t}$ is the distribution of a best response policy $\pi^{\text{BR},t}$ against $\bar{\mu}^t$ when $t \geq 1$. The distribution $\mu^t$ is the distribution of a policy $\bar{\pi}^t$, which is defined as follows for $t \geq 1$:

$$\forall n, \Xi, \qquad \bar{\pi}_n^t(a|x, \Xi) \int_{s=0}^{t} \mu_n^{\text{BR},s}(x|\Xi)ds = \int_{s=0}^{t} \mu_n^{\text{BR},s}(x|\Xi)\pi_n^{\text{BR},s}(a|x, \Xi)ds.$$

**Theorem 2.** *Under the monotony assumption, the exploitability is a strong Lyapunov function of the system for $t \geq 1$: $\frac{d}{dt}\phi(\bar{\pi}^t) \leq -\frac{1}{t}\phi(\bar{\pi}^t)$. Therefore, $\phi(\bar{\pi}^t) = O(\frac{1}{t})$.*

# 6 Experiments with Common Noise

## 6.1 Linear Quadratic Mean Field Game

**Environment:** We use a similar environment as the one described in the Linear Quadratic MFG. On top of the idiosyncratic noise $\epsilon_n$, we add a common noise $\xi_n$, which is assumed to be stationary and i.i.d. We now consider the following dynamics:

$$x_{n+1} = x_n + (K(m_n - x_n) + a_n)\Delta_n + \sigma(\rho\xi_n + \sqrt{1 - \rho^2}\epsilon_n)\sqrt{\Delta_n}.$$

The reward remains unchanged, except that the first moment of the state distribution $\bar{\mu}_n$ now depends on the sequence of common noises $\Xi_n$: $m_n = \mathbb{E}[x_n|\Xi_n]$. We set $\rho = 0.5$.

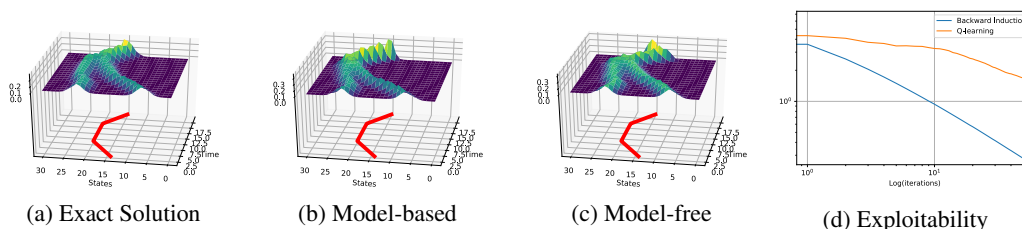

(a) Exact Solution     (b) Model-based     (c) Model-free     (d) Exploitability

Figure 4: Linear Quadratic with Common Noise.

**Numerical results:** On Figure 4, the two separated bell-shaped distributions reassemble and follow the sequence of common noises. Namely, the mean of the distribution moves with the successive common noises, which are represented by the red line below the distribution's evolution. This evolution can be interpreted as a school of fish which undergoes a water flow (*i.e.* the sequence of common noises). Both model-based and model-free approaches approximate the exact solution. The exploitability of model-based still decreases at a rate $O(1/t)$, while the one of model-free decreases more slowly.

## 6.2 The Beach Bar Process

**Environment:** We consider a setting where the bar can close at only one given time step. This gives two possible realizations of the common noise: (1) the bar stays open or (2) it closes at this time step. Here, the dynamics remain unchanged but the reward now depends on the common noise: $r_{open}$ is the same reward as before, whereas $r_{closed}(x_n, a_n, \mu_n) = -\frac{|a_n|}{|\mathcal{X}|} - \log(\mu_n(x_n))$.

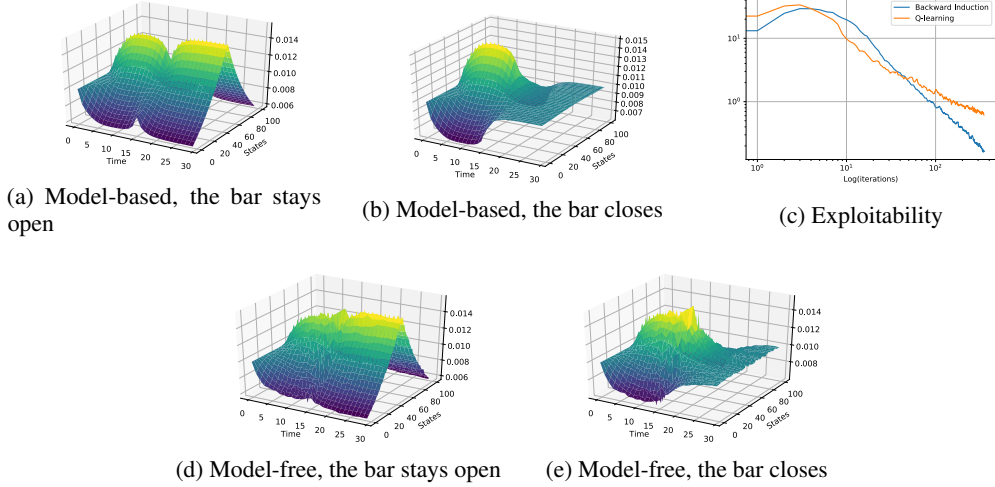

(a) Model-based, the bar stays open

(b) Model-based, the bar closes

(c) Exploitability

(d) Model-free, the bar stays open

(e) Model-free, the bar closes

Figure 5: First Common Noise setting, the bar has a probability $0.5$ of closing at time step $15$.

**Numerical results:** We set the time step of closure at $\frac{N}{2}$ where $N = 30$ is the horizon of the game and the number of states $|\mathcal{X}|$ to $100$. We choose the probability of closure to be $0.5$. Figure 5 shows that the players anticipate the possibility that the bar may close: the density of people next to the bar decreases before the time step of the common noise. After the common noise, the distribution becomes uniform if the bar has closed or people go back next to the bar if the bar stays open. Once again, the exploitability indicates that the model-based and model-free approaches both converge to the Nash equilibrium and that the model-based converges faster.

## 7 Experiment at Scale

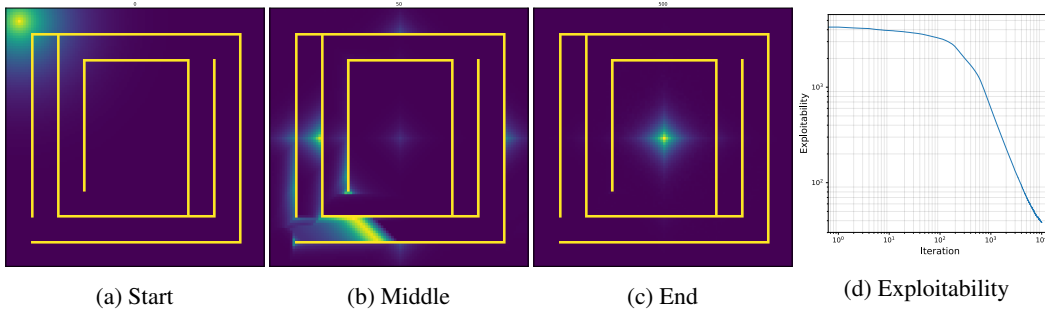

(a) Start

(b) Middle

(c) End

(d) Exploitability

Figure 6: 2D crowd modeling example.

We finally present a crowd modeling experiment, motivated by swarm robotics (see e.g. [88, 113, 48]), where a distribution of players is encouraged to move in a maze towards the center of a $100 \times 100$ grid. The reward at a state $(i, j)$ is described as $r(s = (i, j), a, \mu) = 10 * (1 - \frac{\|(i,j)-(50,50)\|_1}{100}) - \frac{1}{2}\log(\mu(x))$, where the last term captures the aversion for crowded areas. The initial distribution is chosen proportional to $(1 - \frac{\|(i,j)-(5,5)\|_2}{\sqrt{2 \times 95^2}})^{10}$ while being null on the maze obstacles (the yellow strait lines). The evolution of the distribution as well as the exploitability are represented in Figure 6 (a video is available in supplementary material).

## 8    Related Work

**Theoretical results in MFGs:** Theoretical results in terms of uniqueness, existence and stability of Nash equilibrium in such games are numerous, see [30, 19, 36]. A key motivation is that the optimal control derived in an MFG provides an approximate Nash equilibrium in a game with a large but finite number of players. In general, most games are considered in a continuous setting while Gomes *et al.* [62] proved existence results for finite state and action spaces MFGs and [99] considered finite state discounted cost MFGs. An important and challenging extension is the case of players sharing a common source of risk (such as several companies in the same economy market), giving rise to the so-called MFG with common noise, see [37] or [36, Volume II]. These games are usually solved by numerical methods for partial differential equations [9] or probabilistic methods [12, 40, 59].

**Learning in games and MFGs:** The scaling limitations of traditional multi-agent learning methods with respect to the number of players remain quite hard to overcome as the complexity of independent learning methods [56, 96, 95, 109, 92, 58, 57] scales at least linearly with the number of players and some methods may scale exponentially (*e.g.* Nash $Q$-learning [74] or correlated $Q$-learning [66]). By approximating the discrete population by a continuous one, the MFG scheme made learning approaches more suitable and attracted a surge of interest. Model-based methods have been first considered (*e.g.* [116] studied a MF oscillator game, [31] initiated the study of Fictitious Play in MFGs). Recently, several works have focused on model-free methods such as $Q$-learning [68] but the convergence results rely on very strong hypotheses. Note that, although our method can make use of $Q$-learning to learn a best response, it does not rely on it. Also, our method can make use of both model-based and model-free algorithms. Finally, our method relies only on the Lasry-Lions monotonicity condition, which is much less restrictive than a potential or variational structure.

Fictitious Play (FP), which is also a classical method to learn in $N$-player games [97, 93, 71, 73, 72, 96], combined with a model-free algorithm has been considered in [90] but with several inaccuracies, as already pointed out in [111], which focuses on policy gradient methods. However, they study a restricted stationary setting as opposed to the finite time horizon covered by our contribution and their convergence results hold under hardly verifiable assumptions.

Convergence of approximate FP has been proved in [54] (based on the FP analysis of [69]) but without common noise and their analysis is for discrete time FP and only for first-order MFGs (without noise in the dynamics). Our analysis, done in continuous time, is more transparent and works for MFGs with both idiosyncratic and common sources of randomness in the dynamics. Furthermore, their numerical example was stationary whereas we were also able to learn the solution of time-dependent MFGs, which covers a larger scope of meaningful applications. Finally, our analysis provides a rate of convergence ($O(\frac{1}{t})$) while previous FP work in MFG do not.

## 9    Conclusion

In this paper we have shown that Fictitious Play can serve as a basis for building practical algorithms to solve a wide variety of MFGs including finite horizon and $\gamma$-discounted MFGs as well as games perturbed by a common noise. We proved that, in all these settings, the resulting exploitability decreases at a rate of $O(\frac{1}{t})$ and that this metrics can be used to monitor the quality of the control throughout the learning. To illustrate our findings and the versatility of the method, we instantiated the Fictitious Play scheme using Backward Induction and $Q$-Learning to learn intermediate best responses. Application of these instances on different MFGs have shown that the proposed algorithms consistently learned a near-optimal control and led to the desired behaviour for the population of players. This scheme has the potential to scale up dramatically by using advanced reinforcement learning algorithms combined with neural networks for the computation of the best response.

## Broader Impact

**Applications of MFGs:** The MFG model has inspired numerous applications [67] and we hope our work can help practitioners to solve MFGs problems at scale. A popular application focuses on population dynamics modeling [1, 33] including crowd motion modeling [7, 27, 46, 16, 8, 43], opinion dynamics and consensus formation [110, 18, 94], autonomous vehicles [75, 105] or sanitary vaccination [77, 51]. But MFGs have also naturally found applications in banking, finance and economics including banking systemic risk [38, 52], high frequency trading [82, 32], income and wealth distribution [6], economic contract design [53], economics in general [6, 2, 41, 61, 47] or price formation [84, 82, 63]. Energy management or production applications are studied in [10, 44, 50, 17, 79, 85, 67, 5, 42, 65], whereas security and communication applications appear in [89, 100, 70, 115, 80, 81].

**Exploitability as a metric:** One of the leading factor of progress for numerical or learning methods is the clear understanding of which metrics should be optimized. In reinforcement learning, the mean human normalized score is a standard metric of success. In supervised learning, the top 1 accuracy has been the foremost metric of success. We hope the exploitability can achieve such a role on the numerical aspects of MFGs.

## Funding Disclosure

Experiments presented in this paper were carried out using the Grid'5000 testbed, supported by a scientific interest group hosted by Inria and including CNRS, RENATER and several Universities as well as other organizations (see `https://www.grid5000.fr`).

## Footnotes

[2]All the theory can be easily extended in the case where the reward is also time dependent.

[3]In all experiments, we represent $\bar{\mu}$, but applying $\bar{\pi}$ to $\mu_0$ would give the same result as $\bar{\mu} = \mu^{\bar{\pi}}$.

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
