[Supplementary Material · MFG_NeurIPS_supp.pdf]

# A   Continuous Time Fictitious Play in Finite Horizon

In this section, we prove the Fictitious Play convergence result in the absence of common noise. For the sake of clarity, we will write:

$$r^\pi(x,\mu) = \mathbb{E}_{a\sim\pi(.|x)}\left[r(x,a,\mu)\right] \quad \text{and} \quad p^\pi(x'|x) = \mathbb{E}_{a\sim\pi(.|x)}\left[p(x'|x,a)\right]$$

for the rest of this section.

First, we prove the following property, which stems from monotonicity.

**Property 1.** *Let $f$ be a smooth enough function and let assume that the ODE $\dot{\mu} = f(\mu)$ (with $\dot{\mu} = \frac{d}{dt}\mu$) has a solution $(\mu^t)_{t\geq 0} = (\mu_n^t(x))_{t\geq 0, x\in\mathcal{X}}$. If the game is monotone, then:*

$$\sum_{x\in\mathcal{X}} <\nabla_\mu \bar{r}(x,\mu), \dot{\mu}> \dot{\mu}(x) \leq 0.$$

*Proof.* The monotonicity condition implies that, for all $\tau \geq 0$, we have:

$$\sum_{x\in\mathcal{X}}(\mu^t(x) - \mu^{t+\tau}(x))(\bar{r}(x,\mu^t) - \bar{r}(x,\mu^{t+\tau})) \leq 0.$$

Thus:

$$\sum_{x\in\mathcal{X}} \frac{\mu^t(x) - \mu^{t+\tau}(x)}{\tau}\frac{\bar{r}(x,\mu^t) - \bar{r}(x,\mu^{t+\tau})}{\tau} \leq 0.$$

The result follows when $\tau \to 0$. $\qquad\square$

**Property 2.** *Let $\hat{\pi}^t = (\hat{\pi}_n^t)_{n=0,\dots,N}$ be a sequence of time-dependent policies and let $\mu^{\hat{\pi}^t} = (\mu_n^{\hat{\pi}^t}(x))_{n=0,\dots,N,x\in\mathcal{X}}$ be the sequence of their distributions over states. Let us denote, for all $t,n,x$, $\bar{\mu}_n^t(x) = \frac{1}{t}\int_0^t \mu_n^{\hat{\pi}^s}(x)ds$. Then, the policy generating this average distribution is:*

$$\bar{\pi}_n^t(a|x) = \frac{\int_0^t \mu_n^{\hat{\pi}^s}(x)\hat{\pi}_n^s(a|x)ds}{\int_0^t \mu_n^{\hat{\pi}^s}(x)ds}. \tag{1}$$

*Note that $\int_0^t \mu_n^{\hat{\pi}^s}(x)ds$ can be chosen to be strictly positive as one can choose an arbitrary policy on the time interval $[0,1]$ (for example, the uniform policy).*

*Or, more simply, one can write:*

$$\bar{\mu}_n^t(x)\bar{\pi}_n^t(a|x) = \frac{1}{t}\int_0^t \mu_n^{\hat{\pi}^s}(x)\hat{\pi}_n^s(a|x)ds. \tag{2}$$

*Moreover, we have:*

$$\dot{\bar{\mu}}_n^t(x)\bar{\pi}_n^t(a|x) + \bar{\mu}_n^t(x)\dot{\bar{\pi}}_n^t(a|x) = \frac{1}{t}\left[\mu_n^{\hat{\pi}^t}(x)\hat{\pi}^t(a|x) - \bar{\mu}_n^{\hat{\pi}^t}(x)\bar{\pi}_n^t(a|x)\right].$$

*Proof.* Let us start with the following equality, which holds by definition of the dynamics:

$$\mu_{n+1}^{\hat{\pi}^s}(x') = \sum_{x\in\mathcal{X}}\sum_{a\in\mathcal{A}} p(x'|x,a)\hat{\pi}_n^s(a|x)\mu_n^{\hat{\pi}^s}(x).$$

Then, taking on both sides the average over the Fictitious Play time yields:

$$\frac{1}{t}\int_0^t \mu_{n+1}^{\hat{\pi}^s}(x')ds = \sum_{x\in\mathcal{X}}\sum_{a\in\mathcal{A}} p(x'|x,a)\frac{1}{t}\int_0^t \hat{\pi}_n^s(a|x)\mu_n^{\hat{\pi}^s}(x)ds.$$

The left hand side is $\bar{\mu}_{n+1}^t(x')$ by definition, and the time average in the right hand side can be written as:

$$\frac{1}{t}\int_0^t \hat{\pi}_n^s(a|x)\mu_n^{\hat{\pi}^s}(x)ds = \frac{\int_0^t \hat{\pi}_n^s(a|x)\mu_n^{\hat{\pi}^s}(x)ds}{\int_0^t \mu_n^{\hat{\pi}^s}(x)ds}\frac{1}{t}\int_0^t \mu_n^{\hat{\pi}^s}(x)ds = \bar{\mu}_n^{\hat{\pi}^t}(x)\bar{\pi}_n^t(a|x).$$

Combining the terms, we obtain:

$$\bar{\mu}_{n+1}^t(x') = \sum_{x\in\mathcal{X}}\sum_{a\in\mathcal{A}} p(x'|x,a)\bar{\mu}_n^{\hat{\pi}^t}(x)\bar{\pi}_n^t(a|x),$$

which proves that the policy $\bar{\pi}_n^t$ defined in (1) indeed generates $\bar{\mu}_n^t$. The other equalities in the statement can be deduced from here readily. $\qquad\square$

Based on the above properties, we now proceed to the proof of the convergence of Fictitious Play (Theorem 1) in the finite horizon case.

*Proof of Theorem 1.* To alleviate the notation, given a policy $\pi$, we denote:

$$r^\pi(x,\mu) = \sum_a \pi(a|x)r(x,a,\mu).$$

We start by noticing that, thanks to the structure of the reward coming from the monotonicity assumption,

$$\nabla_\mu r^{\pi_n^{\text{BR},t}}(x,\bar{\mu}_n^t) = \nabla_\mu \bar{r}(x,\bar{\mu}_n^t) \text{ and } \nabla_\mu r^{\pi_n}(x,\bar{\mu}_n^t) = \nabla_\mu \bar{r}(x,\bar{\mu}_n^t). \tag{3}$$

Moreover, from Property 2 with $\pi$ replaced by $\hat{\pi}^{\text{BR}}$ and $\mu^{\hat{\pi}^t}$ replaced by $\mu^{\text{BR},t}$, we obtain (2). Dropping the overlines to alleviate the presentation (so $\mu^t$ and $\pi^t$ denote respectively the average sequence of distributions and the average sequence of policies), it implies:

$$\mu_n^t(x)\frac{d}{dt}\pi_n^t(a|x) = \frac{1}{t}\mu_n^{\text{BR},t}(x)[\pi_n^{\text{BR},t}(a|x) - \pi_n^t(a|x)]. \tag{4}$$

Moreover, recall that:

$$\frac{d}{dt}\mu_n^t(x) = \frac{1}{t}\left[\mu_n^{\text{BR},t}(x) - \mu_n^t(x)\right]. \tag{5}$$

From the above observations, we deduce successively:

$$\frac{d}{dt}\phi(\pi^t) = \frac{d}{dt}\left[\max_{\pi'} J(\mu_0,\pi',\mu^t) - J(\mu_0,\pi^t,\mu^t)\right]$$

$$= \sum_{n=0}^N \sum_{x\in\mathcal{X}}\left[ <\nabla_\mu r^{\pi_n^{\text{BR}}}(x,\mu_n^t), \frac{d}{dt}\mu_n^t> \mu_n^{\text{BR},t}(x) - <\nabla_\mu r^{\pi_n}(x,\mu_n^t), \frac{d}{dt}\mu_n^t> \mu_n^t(x)\right.$$

$$\left. - <\frac{d}{dt}\pi_n^t(.|x), r(x,.,\mu_n^t)> \mu_n^t(x) - r^{\pi_n}(x,\mu_n^t)\frac{d}{dt}\mu_n^t(x)\right]$$

$$= \sum_{n=0}^N \sum_{x\in\mathcal{X}}\left[ t<\nabla_\mu \bar{r}(x,\mu_n^t), \frac{d}{dt}\mu_n^t> \frac{1}{t}\left(\mu_n^{\text{BR},t}(x) - \mu_n^t(x)\right)\right]$$

$$+ \sum_{n=0}^N \sum_{x\in\mathcal{X}}\left[\frac{1}{t}r^{\pi_n}(x,\mu_n^t)\mu_n^t(x) - \frac{1}{t}r^{\pi_n^{\text{BR},t}}(x,\mu_n^t)\mu_n^{\text{BR},t}(x)\right]$$

$$= -\frac{1}{t}\phi(\pi^t) + \sum_{n=0}^N \sum_{x\in\mathcal{X}}\left[t<\nabla_\mu \bar{r}(x,\mu_n^t), \frac{d}{dt}\mu_n^t> \frac{d}{dt}\mu_n^t(x)\right],$$

where the third equality holds by (3), (4) and (5). Note that the product $<\nabla_\mu \bar{r}(x,\mu_n^t), \frac{d}{dt}\mu_n^t>$ in the last sum above is non-positive thanks to Property 1 (*i.e.*, thanks to the monotonicity assumption). Hence, the conclusion holds. $\qquad\square$

# B Continuous Time Fictitious Play in Finite Horizon with Common Noise

In this section, we prove the convergence result of continuous time Fictitious Play in finite horizon MFGs with common noise (Theorem 2). The reasoning is similar as in the finite horizon case without common noise (Appx. A). The only difference comes from the conditioning with the common noise.

*Proof of Theorem 2.* For any policy, recall that we write $\pi_{n,\Xi}^t(a|x) = \pi_n^t(a|x,\Xi)$.

We first note that, by the structure of the reward function, we have,

$$\nabla_\mu r^{\pi_{n,\Xi,\xi}^{\mathrm{BR},t}}(x,\mu_{n|\Xi}^t) = \nabla_\mu \bar{r}(x,\mu_{n|\Xi}^t) \text{ and } \nabla_\mu r^{\pi_{n,\Xi,\xi}}(x,\mu_{n|\Xi}^t) = \nabla_\mu \bar{r}(x,\mu_{n|\Xi}^t).$$

Moreover,

$$- < \frac{d}{dt}\pi_{n,\Xi,\xi}^t(.|x), r(x,.,\mu_{n|\Xi}^t) > \mu_{n|\Xi}^t(x) = -\frac{1}{t}r^{\pi_{n,\Xi,\xi}^{\mathrm{BR},t}}(x,\mu_{n|\Xi}^t)\mu_{n|\Xi}^{\mathrm{BR},t}(x) + \frac{1}{t}r^{\pi_{n,\Xi,\xi}}(x,\mu_{n|\Xi}^t)\mu_{n|\Xi}^{\mathrm{BR},t}(x)$$

and

$$-r^{\pi_{n,\Xi,\xi}}(x,\mu_{n|\Xi}^t)\frac{d}{dt}\mu_{n|\Xi}^t(x) = \frac{1}{t}r^{\pi_{n,\Xi,\xi}}(x,\mu_{n|\Xi}^t)\mu_{n|\Xi}^t(x) - \frac{1}{t}r^{\pi_{n,\Xi,\xi}}(x,\mu_{n|\Xi}^t)\mu_n^{\mathrm{BR},t}(x)$$

Using the definition of exploitability together with the above remarks, we deduce:

$$\frac{d}{dt}\phi(\pi^t) = \frac{d}{dt}\left[\max_{\pi'} J(\mu_0,\pi',\mu^\pi) - J(\mu_0,\pi,\mu^\pi)\right]$$

$$= \sum_{n=0}^{N} \sum_{\Xi,|\Xi|=n} \sum_{\xi} P(\Xi.\xi) \sum_{x \in \mathcal{X}} \Big[ < \nabla_\mu r^{\pi_{n,\Xi,\xi}^{\mathrm{BR},\xi}}(x,\mu_{n|\Xi}^t), \frac{d}{dt}\mu_{n|\Xi}^t > \mu_{n,\Xi,\xi}^{\mathrm{BR},t}(x)$$

$$- < \nabla_\mu r^{\pi_{n,\Xi,\xi}}(x,\mu_{n|\Xi}^t,\xi), \frac{d}{dt}\mu_{n|\Xi}^t > \mu_{n|\Xi}^t(x)$$

$$- < \frac{d}{dt}\pi_{n,\Xi,\xi}^t(.|x), r(x,.,\mu_{n|\Xi}^t) > \mu_{n|\Xi}^t(x) - r^{\pi_{n,\Xi,\xi}}(x,\mu_{n|\Xi}^t)\frac{d}{dt}\mu_{n|\Xi}^t(x)\Big]$$

$$= \sum_{n=0}^{N} \sum_{\Xi,|\Xi|=n} \sum_{\xi} P(\Xi.\xi) \sum_{x \in \mathcal{X}} \Big[ t < \nabla_\mu \bar{r}(x,\mu_{n|\Xi}^t)), \frac{d}{dt}\mu_{n|\Xi}^t > \frac{1}{t}\left(\mu_{n|\Xi}^{\mathrm{BR},t}(x) - \mu_{n|\Xi}^t(x)\right)\Big]$$

$$+ \sum_{n=0}^{N} \sum_{\Xi,|\Xi|=n} \sum_{\xi} P(\Xi.\xi) \sum_{x \in \mathcal{X}} \Big[\frac{1}{t}r^{\pi_{n,\Xi,\xi}}(x,\mu_{n|\Xi}^t)\mu_{n|\Xi}^t(x) - \frac{1}{t}r^{\pi_{n,\Xi,\xi}^{\mathrm{BR},t}}(x,\mu_{n|\Xi}^t)\mu_{n|\Xi}^{\mathrm{BR},t}(x)\Big]$$

$$= -\frac{1}{t}\phi(\pi^t) + \sum_{n=0}^{N} \sum_{\Xi,|\Xi|=n} \sum_{\xi} P(\Xi.\xi) \sum_{x \in \mathcal{X}} \Big[ t < \nabla_\mu \bar{r}(x,\mu_{n|\Xi}^t), \frac{d}{dt}\mu_{n|\Xi}^t > \frac{d}{dt}\mu_{n|\Xi}^t(x)\Big],$$

where the last term is non-positive by Property 1 (*i.e.*, thanks to the monotonicity assumption).  □

**Experiments: A More Complex Setting for the Beach Bar Process with common noise**

**Environment:** Following the first setting of the paper where the bar could only close at one given time step, we now introduce a second more complex setting, bringing also of common noise in the beach bar process. Namely, the bar has a probability $p$ to close at every time step up to a point (in practice, this point is half of the horizon: $\frac{N}{2}$). Once the bar is closed, it does not open again. This setting gives $\frac{N}{2} + 1$ possible realizations of the common noise: (1) the case where the bar never closes and (2) the $\frac{N}{2}$ cases where it closes at any of the first $\frac{N}{2}$ time steps. For the sake of clarity, we only present the evolution of the distributions when the bar finally remains open after $\frac{N}{2}$ time steps, and when it closes at the $\frac{N}{2}^{\text{th}}$ time step.

**Numerical results**: Similarly to the first setting, we take $|\mathcal{X}| = 100$ states and $N = 30$ time steps. As the bar has a probability $p = 0.5$ to close at every time step until $\frac{N}{2}$, the distribution is flatter to anticipate the fact that people might need to spread. We can see that both model-based and model-free approaches converge to a Nash equilibrium and that model-based converges faster than model-free.

(a) Model-based, the bar stays open

(b) Model-based, the bar closes

(c) Exploitability

(d) Model-free, the bar stays open

(e) Model-free, the bar closes

Figure 7: $2^{nd}$ common noise setting, the bar has a probability $p = 0.5$ to close at every time step before $\frac{N}{2}$.

## C  Continuous Time Fictitious Play: the $\gamma$-discounted case

Surprisingly, the analysis also holds in the $\gamma$-discounted case with again the same style of reasoning. However, the distribution considered will be the $\gamma$-weighted occupancy measure instead of the distribution over states. In this section, we reintroduce the notations and we prove similar continuous time FP convergence results.

Consider, given the following:

- a finite state space $\mathcal{X}$ ($x \in \mathcal{X}$),
- a finite action space $\mathcal{A}$ ($a \in \mathcal{A}$),
- the set of distributions over state is $\Delta\mathcal{X}$ ($\mu \in \Delta\mathcal{X}$),
- a reward function $r(x, a, \mu)$,
- the transition function $p(x'|x, a)$,
- a policy: $\pi(a|x)$.

We will write:

- $p^\pi(x'|x) = \mathbb{E}_{a \sim \pi(.|x)}[p(x'|x, a)]$,
- $r^\pi(x, \mu) = \mathbb{E}_{a \sim \pi(.|x)}[r(x, a, \mu)]$,

The cumulative $\gamma$-discounted reward is defined as:

$$J_\gamma(x_0, \pi, \mu) = \mathbb{E}\left[\sum_{n=0}^{+\infty} \gamma^n r(x_n, a_n, \mu) \mid x_{n+1} \sim p(.|x_n, a_n),\ a_n \sim \pi(.|x_n)\right]$$

**Useful properties:** We have $\mu_\gamma^\pi(x') = \mu_0(x') + \gamma \sum_{x \in \mathcal{X}} p^\pi(x'|x)\mu_\gamma^\pi(x)$ (in vectorial notations $\mu_\gamma^{\pi\top} = \mu_0^\top (I - \gamma P^\pi)^{-1}$).
The $\gamma$-discounted reward can be written as: $J_\gamma(x_0, \pi, \mu) = \sum_{x \in \mathcal{X}} \mu_\gamma^\pi(x) r^\pi(x, \mu)$.
We then have a similar formula for the policy generating the average distribution $\bar\mu_\gamma^\pi(x, t) = \frac{1}{t}\int_0^t \mu_\gamma^\pi(x, s)ds$ can be written $\bar\pi_\gamma(a|x, t) = \frac{\int_0^t \mu_\gamma^\pi(x,s)\pi(a|x,s)ds}{\int_0^t \mu_\gamma^\pi(x,s)ds}$.

Finally, we can write:

$$\bar\mu_\gamma^\pi(x, t)\bar\pi_\gamma(a|x, t) = \frac{1}{t}\int_0^t \mu_\gamma^\pi(x, s)\pi(a|x, s)ds$$

And:

$$\dot{\bar\mu}_\gamma^\pi(x, t)\bar\pi_\gamma(a|x, t) + \bar\mu_\gamma^\pi(x, t)\dot{\bar\pi}_\gamma(a|x, t) = \frac{1}{t}\left[\mu_\gamma^\pi(x, t)\pi(a|x, t) - \bar\mu_\gamma^\pi(x, t)\bar\pi_\gamma(a|x, t)\right]. \quad (6)$$

**Fictitious Play in MFGs:** In the $\gamma$-discounted case, Fictitious Play can be written as (for $t \geq 1$):

$$\dot\mu(x, t) = \frac{1}{t}(\mu_\gamma^{\mathrm{BR}}(x, t) - \mu(x, t))$$

where $\mu_\gamma^{\mathrm{BR}}(x, t)$ is the distribution of a best response against $\mu(x, t)$ of policy $\pi^{\mathrm{BR}}(a|x, t)$. In this section, we will write $\pi(a|x, t)$ the policy of the distribution $\mu(x, t)$. From Eq.(6), we can deduce the following property:

**Property 3.**

$$\forall n,\ \dot\pi(a|x, t)\mu(x, t) = \frac{1}{t}\mu_\gamma^{BR}(x, t)[\pi^{BR}(a|x, t) - \pi(a|x, t)]$$

*Proof.* Such representation directly follows from Eq.(6). $\qquad\square$

We are now in position to turn to the Lyapounov congerging property of the Fictitious process.

**Property 4.** *Under the monotony assumption, we can show that the exploitability ($\phi(t) = \max_{\pi'} J_\gamma(x_0, \pi', \mu^\pi) - J_\gamma(x_0, \pi, \mu^\pi)$) is a strong Lyapunov function of the system:*

$$\dot{\phi}(t) \leq -\frac{1}{t}\phi(t)$$

*Proof.*

$$\dot{\phi}(t)$$

$$= \sum_{x \in \mathcal{X}} \Big[ \overbrace{< \nabla_\mu r^{\pi^{\mathrm{BR}}}(x, \mu(t)), \dot{\mu}(t) > \mu_\gamma^{\mathrm{BR}}(x,t) - < \nabla_\mu r^\pi(x, \mu(t)), \dot{\mu}(t) > \mu(x,t)}^{\text{With } \nabla_\mu r^{\pi^{\mathrm{BR}}}(x,\mu(t)) = \nabla_\mu \bar{r}(x,\mu(t)) \text{ and } \nabla_\mu r^\pi(x,\mu(t)) = \nabla_\mu \bar{r}(x,\mu(t))}$$

$$\underbrace{- < \dot{\pi}(.|x,t), r(x,.,\mu(t)) > \mu(x,t)}_{= -\frac{1}{t}r^{\pi^{\mathrm{BR}}}(x,\mu(t))\mu_\gamma^{\mathrm{BR}}(x,t) + \frac{1}{t}r^\pi(x,\mu(t))\mu_\gamma^{\mathrm{BR}}(x,t)} \quad \underbrace{-r^\pi(x,\mu(t))\dot{\mu}(x,t)}_{= \frac{1}{t}r^\pi(x,\mu(t))\mu(x,t) - \frac{1}{t}r^\pi(x,\mu(t))\mu_\gamma^{\mathrm{BR}}(x,t)} \Big]$$

$$= \sum_{x \in \mathcal{X}} \Big[ t < \nabla_\mu \bar{r}(x, \mu(t))), \dot{\mu}(t) > [\frac{1}{t}(\mu_\gamma^{\mathrm{BR}}(x,t) - \mu(x,t))]] $$

$$+ \sum_{x \in \mathcal{X}} \Big[ \frac{1}{t}r^\pi(x,\mu(t))\mu(x,t) - \frac{1}{t}r^{\pi^{\mathrm{BR}}}(x,\mu(t))\mu_\gamma^{\mathrm{BR}}(x,t) \Big]$$

$$= -\frac{1}{t}\phi(t) + \underbrace{\sum_{x \in \mathcal{X}} \Big[ t < \nabla_\mu \bar{r}(x, \mu(t))), \dot{\mu}(t) > \dot{\mu}(x,t)) \Big]}_{\leq 0 \text{ by monotony}}$$

$$\leq -\frac{1}{t}\phi(t)$$

□

**Experiment: the Beach Bar Process with $\gamma$-discounted reward.**

**Environment:** We implement the beach bar process in the $\gamma$-discounted setting.

**Numerical results:** We set $\gamma = 0.9$. The algorithm estimating the best response to a fixed distribution $\mu$ is Policy Iteration in the case of the model based approach and $Q$-learning in the model-free. As the flow of distributions converges towards the stationary distribution which is not time-dependant, we only plot the final distribution obtained after 300 time steps (and not the evolution throughout time as before). In particular, we notice that model-based and model-free approaches converge towards the same distribution. We can also observe that the convergence rate of exploitability is $O(1/t)$ for the model-based and slower for the model-free approach.

(a) Model-based        (b) Model-free        (c) Exploitability

Figure 8: Final distributions and exploitability in the $\gamma$-discounted case

## D Algorithms

---

**Algorithm 2:** $Q$-Learning in Mean Field Games

---

**input** : Start with a fixed distribution $\mu = (\mu_k)_k$ and $Q^k = 0$ and $\epsilon$ and the learning rate $\alpha$.

1 **for** $k = 0, \ldots, K$*:* **do**

2     sample $x_0^k \sim \mu_0$ ;

3     **for** $n = 0, \ldots, N$*:* **do**

4         $a_n^k$ is $\epsilon$-greedy with respect to $Q^k(x_n^k, .)$.;

5         if not terminal sample $x_{n+1}^k$ according to $p(.|x_n^k a_n^k)$.;

6         $Q_n^{k+1}(x_n^k, a_n^k) = (1 - \alpha)Q_n^{k+1}(x_n^k, a_n^k) + \alpha[r(x_n^k, a_n^k, \mu_{k-1}) + \max_b Q_{n+1}^k(x_{n+1}^k, b)]$;

7 **return** $\pi^*$ *a greedy policy with respect to* $Q^K$

---

**Algorithm 3:** Empirical Density Estimation

---

**input** : Start with a fixed policy $\pi$ and an initial distribution $\mu_0 = \mu_0^\pi$

1 **for** $k = 0, \ldots, K$*:* **do**

2     sample $x_0^k \sim \mu_0$ ;

3     **for** $n = 0, \ldots, N$*:* **do**

4         $a_n^k$ with respect to $Q^k(x_n^k, .)$.;

5         if not terminal sample $x_{n+1}^k$ according to $p(.|x_n^k a_n^k)$.

6 Finally $\forall x, n \in \mathcal{X} \times \{0, \ldots, N\} \; \hat{\mu}_n^\pi(x) = \frac{1}{K+1} \sum_{k=0}^{K} 1_{x_n^k = x}$;

7 **return** $\hat{\mu}_n^\pi$

---

**Algorithm 4:** Backward Induction in Mean Field Games

---

**input** : Start with a fixed distribution $\mu = (\mu_k)_k$ and a terminal $Q$-function
        $Q_N^\mu(x, a) = r(x, a, \mu_N)$

1 **for** $n = N, \ldots, 0$*:* **do**

2     $\pi_k^*$ is greedy with respect to $Q_k^\mu(x, a)$. ;

3     $\forall a, x \in \mathcal{A} \times \mathcal{X} \; Q_{n-1}^\mu(x, a) = r(x, a, \mu_{n-1}) + \sum_{x' \in \mathcal{X}} p(x'|x, a) \max_b Q_n^\mu(x', b)$ ;

4 **return** $\pi^*$

---

**Algorithm 5:** Density Estimation

---

**input** : Start with a fixed policy $\pi$ and an initial distribution $\mu_0 = \mu_0^\pi$

1 **for** $n = 1, \ldots, N$*:* **do**

2     $\forall x \in \mathcal{X} \; \mu_n^\pi(x') = \sum_{x, a \in \mathcal{X} \times \mathcal{A}} \pi_{n-1}(a|x) p(x'|x, a) \mu_{n-1}^\pi(x)$ ;

3 **return** $\mu^\pi$

---

# E Linear Quadratic Model

## E.1 Description

For the sake of completeness, we explain here how we obtained the benchmarck solution for the LQ problem. The original model has been introduced in [38] and corresponds to the continuous time and continuous spaces version of the LQ problem implemented in Section 4. Each player can influence their speed with a control denoted by $\alpha_t$. The dynamics of the players is linear in their state, their control and the mean position, denoted by $\bar{m}_t$. It is affected by an idiosyncratic source of randomness $\mathbf{W} = (W_t)_{t\geq 0}$ as well as a common noise in the form of a Brownian motion $\mathbf{W}^0 = (W_t^0)_{t\geq 0}$. Given a flow of *conditional* mean positions $\bar{\boldsymbol{\mu}} = (\bar{\mu}_t)_{t\in[0,T]}$ adapted to the filtration generated by $\mathbf{W}^0$, the cost function of a representative player is defined as:

$$J(a; \bar{\boldsymbol{\mu}}) = \mathbb{E}\left[\int_0^T \left(\frac{1}{2}a_t^2 - qa_t(\bar{\mu}_t - X_t) + \frac{\kappa}{2}(\bar{\mu}_t - X_t)^2\right) dt + \frac{c_{\text{term}}}{2}(\bar{\mu}_T - X_T)^2\right]$$

Subject to the dynamics:

$$dX_t = [K(\bar{\mu}_t - X_t) + a_t]dt + \sigma\left(\rho\, dW_t^0 + \sqrt{1-\rho^2}dW_t\right).$$

At equilibrium, we must have $\bar{\mu}_t = \mathbb{E}[X_t|(W_s^0)_{s\leq t}]$ for every $t \in [0,T]$.

Here, $\rho \in [0,1]$ is a constant parameterizing the correlation between the noises, and $q, \kappa, c, a, \sigma$ are positive constants. We assume that $q \leq \kappa^2$ so that the running cost is jointly convex in the state and the control variables.

The terms $(\bar{\mu}_t - X_t)$ in the dynamics and the cost function attract the process towards the mean $\bar{\mu}_t$. For the interpretation of this model in terms of systemic risk, the reader is referred to [38]. The model is of linear-quadratic type and has an explicit solution through a Riccati equation, which we use as a benchmark. The optimal control at time $t$ is a linear combination of $X_t$ and $\bar{\mu}_t$, whose coefficients depend on time. More precisely, it is given by:

$$a_t = (q + \eta_t)(\bar{\mu}_t - X_t),$$

where $\eta$ solves the following Riccati ODE:

$$\dot{\eta}_t = 2(a+q)\eta_t + \eta_t^2 - (\kappa - q^2), \qquad \eta_T = c_{\text{term}},$$

whose solution is explicitly given by:

$$\eta_t = \frac{-(\kappa - q^2)\left(e^{(\delta^+ - \delta^-)(T-t)} - 1\right) - c\left(\delta^+ e^{(\delta^+ - \delta^-)(T-t)} - \delta^-\right)}{\left(\delta^- e^{(\delta^+ - \delta^-)(T-t)} - \delta^+\right) - c\left(e^{(\delta^+ - \delta^-)(T-t)} - 1\right)}$$

where $\delta^{\pm} = -(a+q) \pm \sqrt{R}$ with $R = (a+q)^2 + (\kappa - q^2) > 0$.

# F  Common Success Metrics in Mean Field Games

The optimal value function satisfies the recursive equation:

$$V_N^{*,\mu}(x) = r(x, \mu_N), \qquad V_{n-1}^{*,\mu}(x) = \max_a \left\{ r(x, a, \mu_{n-1}) + \sum_{x' \in \mathcal{X}} p(x'|x, a) V_n^{*,\mu}(x') \right\}.$$

In particular, by definition:

$$\max_{\pi'} J(\mu_0, \pi', \mu^\pi) = \mathbb{E}_{x \sim \mu_0}[V_0^{*,\mu^\pi}(x)]$$

And:

$$J(\mu_0, \pi, \mu^\pi) = \mathbb{E}_{x \sim \mu_0}[V_0^{\pi,\mu^\pi}(x)].$$

Let $(x, \mu) \mapsto a^*(x, \mu)$ be such that for every $n$ and (reasonable?) $\mu$:

$$V_{n-1}^{*,\mu}(x) = r(x, a^*(x, \mu_{n-1}), \mu_{n-1}) + \sum_{x' \in \mathcal{X}} p(x'|x, a^*(x, \mu_{n-1})) V_n^{*,\mu}(x'), \qquad (7)$$

*i.e.* $a^*$ is an optimal control. Then, one way to check whether the value function we learned (*e.g.* deduced from the $Q$-table) is a good approximate solution, is to compute the residual in the fixed point equation (7). In other words, if the learned value function is $\tilde{V}$ and the policy is $\pi$ with associated distribution $\mu^\pi$, then, we compute:

$$\tilde{V}_{n-1}(x) - \left[ r(x, a^*(x, \mu_{n-1}^\pi), \mu_{n-1}^\pi) + \sum_{x' \in \mathcal{X}} p(x'|x, a^*(x, \mu_{n-1}^\pi)) \tilde{V}_n(x') \right]$$

for every $n, x$. Taking the norm over $(n, x) \in \{1, \ldots, N\} \times \mathcal{X}$ provides a metric to assess the convergence of the value function.

**Link with fixed-point iterations:** One of the most basic methods to compute a MFG equilibrium is to iteratively solve the forward equation for the distribution and the backward equation for the value function. A typical stopping criterion is that the distribution and the value function do not change too much between two successive iterations. We argue that this property implies an upper bound on the exploitability. To be specific, say that at iteration $k$, given a value function $V_k$ and its associated optimal control $\pi_k$, we compute the induced flow of distributions $\mu_k = \mu^{\pi_k}$, and then we compute the value function $V_{k+1}$ and the best response $\pi_{k+1}$ of an infinitesimal player against this flow of distributions. Note that:

$$\max_{\pi'} J(\mu_0, \pi', \mu_k) = \max_{\pi'} J(\mu_0, \pi', \mu^{\pi_k}) = J(\mu_0, \pi_{k+1}, \mu^{\pi_k}) = \sum_x V_{k+1,0}(x) \mu_0(x)$$

And:

$$J(\mu_0, \pi_k, \mu_k) = \sum_x V_{k,0}(x) \mu_0(x).$$

Hence, if we know that $\|V_{k+1} - V_k\|_\infty := \sup_{x,n} |V_{k+1,n}(x) V_{k,n}(x)| < \epsilon$, then, in particular, $|V_{k+1,0}(x) - V_{k,0}(x)| < \epsilon$ for all $x$ and hence the exploitability is at most $\epsilon$ too. Conversely, under suitable regularity assumptions, we can expect that a small exploitability implies $V_{k+1} \approx V_k$ not only at time 0 but at every time.