[Reviews · NeurIPS 2020]

Review 1

Summary and Contributions: This paper studies continuous-time fictitious play (CTFP) in mean-field games (MFGs), and shows that the exploitability metric (which characterizes the quality of an approximate Nash equilibrium) converges to 0 at a rate of O(1/t) for various settings including both finite horizon (with and without common noises) and discounted infinite horizon cases. The authors also numerically tested two discrete time implementation of CTFP, one with Backward Induction and the other with Q-learning, and show that these algorithms converge as expected to an interpretable population state distribution on various MFG examples (discretized linear-quadratic, beach bar with and without common noises, and maze). The major contribution is to establish the convergence rate of CTFP on generic MFGs, and to numerical test the performance of two discrete-time algorithms motivated by fictitious play.

Strengths: The major strengths of this paper are listed below: 1. It establishes the convergence rate of CTFP for generic MFGs with mild assumptions (monotonicity), and the results apply to both finite horizon MFGs (with and without common noises) and infinite horizon discounted MFGs. 2. It provides rich numerical tests of two discrete-time instantiations of CTFP, one with Backward Induction and the other with Q-learning, on a wide range of examples. The empirical findings mostly match the theoretical results and the interpretations (e.g., those on the learned population state distributions) are inspiring.

Weaknesses: The major weaknesses of this paper are listed here. More details can be found below. 1. The presentation of the CTFP theory is confusing and inconsistent. In particular, Section 3 needs some substantial improvement. 2. There is a gap between the CTFP theory and the practical algorithms based on Backward Induction and Q-learning, and the details about the two algorithms are not clearly described and explained in the main text. And although some ingredients of the algorithms are written with more details in the appendix, it’s still unclear how each of the algorithm work as a whole. Below we make some more detailed comments on the aforementioned points (and beyond): 1. In Section 2, the authors introduce two discrete time instantiations of the CTFP, one with Backward Induction and the other with Q-learning. However, the description of these algorithms are too brief in the main text, which makes it hard to understand what is tested in the various numerical experiments, especially for general readers (of the NeurIPS community, for example). Moreover, although the best response subproblem and the population distribution calculation are described in more details in the Appendix (Algorithms 2, 3, 4, 5), the authors do not provide sufficient details on how they are put together into two algorithms (that are numerically tested). For example, how is the obtained \hat{\mu}_n^{\pi} in Algorithm 3 used in Algorithm 2? Also, in Algorithm 2, the input states that “Start with a fixed distribution \mu = and …”: what is the \mu equal to? And what is the relationship among \mu, \mu_0, \mu_{k-1} and \hat{\mu}_n^{\pi}? 2. Again for the two practical algorithms, I’m wondering why the authors do not employ the randomization strategy of the policy selection (uniformly over previously obtained policies) and the incremental averaging of the mean field as is done in [51]. The authors may want to add some explanations of this. In addition, I noticed that the practical algorithms use greedy policies after obtaining the Q-values. However, as noted in [64] and [Angiuli, A., Fouque, J. P., & Laurière, M. (2020). Unified Reinforcement Q-Learning for Mean Field Game and Control Problems. arXiv preprint arXiv:2006.13912], some exploration trade-off (using soft-max operator or \epsilon-greedy strategies) should be adopted to avoid instability in the learning process in general, especially in the finite state-action spaces setting (as considered in the current paper). The authors may want to double check this and should otherwise provide explanations on why such a randomization is unnecessary in their numerical examples. By the way, notice that in fact for the examples in Section 4.1 and 7, the exploitability never goes below 10^0, which does not really indicate convergence. So the authors may want to double check whether adding some soft-max or epsilon-greedy smoothing in defining the next policy \pi from the learned Q-values in Algorithms 2 and 4 will improve the convergence. In any case, such kind of inexactness or randomization is not covered in the idealistic CTFP theory studied in this paper, which limits the contribution of the work. 3. On a related point, no (new) theory is provided for the two practical algorithms, leaving a non-negligible gap between the theory and practice in this paper. This could be partly resolved by considering (vanishing) inexactness or perturbation errors in CTFP and establish the convergence (rate) theory accordingly. 4. In Section 3, the authors fail to make a smooth transition from the discrete time fictitious play for MFGs at the end of Section 2 to the continuous-time setting. It would be very helpful to clearly specify how the authors move from the updates in Algorithm 1 to the CTFP at the beginning of Section 3. Also, although t=1 is also used as “watershed” in [68], which studies (mostly) the same CTFP dynamics, the authors should explain why they need to initialize with t<1 and then state the results only for t\geq 1. In particular, is the choice of 1 essential? And why can’t we start with t=0 and derive results for all t>0? 5. Again in Section 3, the presentation of the CTFP setting and analysis is hard to follow, and contains several inconsistencies. In particular, the best response policy is not formally defined, and the authors first state that \pi_n^{BR,t} is a best response policy (line 104) and then state that it is chosen arbitrarily for t<=1 (line 107), which seems to be a bit inconsistent. In addition, it is also quite unclear why it matters to introduce the equation involving \pi_n^t before stating Theorem 1. I checked Appendix A and find it much easier to follow. The authors should more carefully rewrite Section 3 with necessary details in Appendix A to make it easier to understand (without looking at the whole Appendix A). 6. In Section 5, the assumption on the common noise sequence is not clearly made. Are they allowed to be arbitrary (e.g., can they be history dependent/not even Markovian), or do they have to be stationary and i.i.d.? This is important especially for understanding the numerical experiments in Section 6 (and in particular why they are different and potentially more nontrivial than the previous experiments without common noises). To make it better, the authors should also specify the choice of the \xi_n sequence in Sections 6.1 and 6.2 with details, so that the readers can easily understand the difference. Otherwise, for Section 6.1, the readers may wonder why one cannot regard \rho\xi_n+\sqrt{1-\rho^2}\epsilon_n as a dummy placeholder for the original \epsilon_n term (without common noises), with no essential difference. In general, without a clear explanation of the common noise sequence, readers may feel that this is no more than an MFG setting with non-stationary transition probabilities and random non-stationary rewards.

Correctness: Yes. I think the claims and method are correct and the empirical methodology is also correct as far as understand.

Clarity: Yes, mostly, apart from the drawbacks mentioned above in the weaknesses section.

Relation to Prior Work: Yes, mostly. I would suggest the authors to compare more explicitly with the CTFP literature (e.g, [68, 87]) and state more clearly what is the difference in the setting (referred by the authors as “simpler games”) considered in the literature and the setting considered in this paper. See the additional feedback for more comments related to this point.

Reproducibility: No

Additional Feedback: ================== post rebuttal ==================================== I have read the authors' rebuttal and the other reviews. I think the authors have showed sufficient effort in addressing most of the raised questions, and I'm satisfied with most of their explanations (pending that they do make the edits as they promised in the rebuttal). So I decide to raise my score from 5 to 6. These being said, I still have some small concerns and confusion. Firstly, the authors mentioned in the rebuttal that "As we use \bar{\mu}^j, we don’t need to select the policy uniformly over previously obtained policies". However, in Algorithm 1, an averaged policy \bar{\pi}_j is still computed and used. Secondly, the authors mentioned that "We also introduce a new theory of common noise for the two practical algs". However, the theory of common noise CTFP is not related to the practical (discrete time) algorithms, if I understand correctly. In general, the authors may want to be more careful in their revision when they want to add such kind of additional comments, which may introduce new confusion to the readers. Finally, regarding the common noise setting, I noticed that one of the key differences is that the policies (and mean fields) are dependent on the noise sequence. So in order to apply these polices, the realization of the noise sequence has to be known a priori. In reality, how do people know the noise sequence in advance (for example, whether the bar is closed or open at some time step in the example in Section 6.2)? The authors should add some comments on the applicability of the common noise policies in reality. ====================================================== In general, I think the convergence rate result of CTFP is clean and good, and numerical experiments are thorough. However, as mentioned above, the presentation of the CTFP theory (the central result) needs substantial improvement, and there is a non-negligible gap between the theory and the numerical experiments. So I would prefer to provide a more conservative rating at present, but the authors are welcome to argue and explain more to try to address my concerns in the rebuttal. Last but not least, please find some suggestions on typo fixing, writing improvement and so on below: 1. Appendix F seems to be separated from the other parts of the paper, including both the main text and the appendix. The authors may want to connect it with the main text or other parts of the appendix to make it easier to understand the thesis of this section. 2. In line 38, “see Appx. E” should be “see Appx. C”. Similarly, in line 137, “see Appx. C” should be “see Appx. E”. 3. In line 39, the authors may want to explain more about what they mean exactly by “simpler games”. 4. In lines 41-42, the authors claim that they derive the “first time convergence properties … MFGs with common noises”. Do you mean that there have been existing results for CTFP on MFGs without common noises (which is the main result of Section 3)? If not, the authors may want to state this more clearly. 5. In line 55, the authors assume that the state and action spaces are finite. But in a closely related paper [51], such finiteness assumptions are not made. The authors may want to comment and explain more on why this is needed and if this is essential to the proofs and so on. 6. In Algorithm 1, line 4, what is the exploitability e_{j-1} used for? it does not seem to be used anywhere in the algorithm. Is it just used to determine whether to terminate the algorithm? 7. In line 106, “proof in A” should better be “proof in Appx. A”. 8. In Theorem 1, “… of the system, $\forall t\geq 1$ …” should better be “… of the system, \textit{i.e.}, $\forall t\geq 1$ …”. 9. In Section 4.1, the quantities \Delta_n and \bar{\mu}_n are not explained until about 6 lines after their introduction, which should be put forward. Also, what does “nodes” mean in “… move up to M nodes …”? 10. In Section 4.2, numerical results, the authors may want to specify the bar location chosen so that the readers can interpret the plots even better. 11. In Section 5, there are several notational inconsistencies in terms of \mu_n(x|\Xi). The authors sometimes use \mu_0(x,\Xi_0) and \mu_n(x,\Xi), and sometimes even write \mu_{n+1}^{\pi}(x’|\Xi.\xi) and so on. These should be unified. 12. In line 218, “dtistribution” should be “distribution”. 13. In line 257, “metrics” should be “metric”. 14. In line 260, “Application” should be “Applications”. 15. In the appendix, Algorithms 2 and 3, “p(\dot|x_n^ka_n^k)” should be “p(\cdot|x_n^k,a_n^k)”. Also, references [65] and [66] are the same and one of them should be removed.


Review 2

Summary and Contributions: The authors study a continuous-time Fictitious Play learning algorithms into discrete state mean-field games. A common noise, i.e. stochastic perturbation in probability densities, is considered. They connect learning in games with mean-field games. They also design several numerical experiments in either model-based or model-free settings.

Strengths: The author introduces the common noise from Mean field games into Fictitious Play. There are two potential benefits of the proposed approach. On the one hand, they apply noises in the probability density of players, which may have potential advantages in reinforcement learning problems. On the other hand, the connection between Mean field games and reinforcement learning is fruitful. The mean-field game also introduces physics-informed dynamics or the Markov process. A particular example is a Fokker-Planck equation. In this perspective, this connection will be new directions for future studies.

Weaknesses: The authors empahize examples in Mean field games. Is it possible to demonstrate several classical reinforcement learning examples and illustrate the connection?

Correctness: The method is correct.

Clarity: The paper is well written.

Relation to Prior Work: There are also several machine learning works in mean field games for primal dual algorithms, which are relevant. Lin et.al. APAC-Net: Alternating the Population and Agent Control via Two Neural Networks to Solve High-Dimensional Stochastic Mean Field Games. Lars. et.al. A machine learning framework for solving high-dimensional mean field game and mean field control problems.

Reproducibility: Yes

Additional Feedback: The paper is an attempt to connect the mean-field game with reinforcement learning. The examples here are mainly from classical mean-field games. In other words, the game dynamics are at a continuous level, which consists of the Fokker-Planck equation and the Hamilton-Jacobi equation. Generalizations of these game dynamics to a broader reinforcement setting will be interesting directions. It may be important to consider some classical reinforcement learning examples with their connection with the MFG theory. I have read all the response. The authors address my concerns. A good paper.


Review 3

Summary and Contributions: The paper studies the performance of a continous time fictitious play algorithm in the context of finite (or discounted infinite) horizon Mean Field Games (MFG). Such games have found increasing use in modeling a variety of context, from oligopoly dynamics, advertising and sponsored search markets, ridesharing and two-sided platform design. Given their widespread use in modeling, a learning algorithm for finding the Nash equilibrium in such games has important practical implications. Under some monotonicity conditions on the reward functions of the agents, the authors show that in the continuous time fictitious play algorithm, the averaged sequence of policies converges to the unique Nash equilibrium, at rate O(1/t). This is shown by exhibiting that an exploitability metric is a strong Lyapunov function for the learning dynamics. The authors illustrate their analytical results through a set of numerical examples, including one with common noise affecting all agents' state transitions. After author response: I read through the author response. Regarding the response to (3), I agree that most convergence rate results for fictitious play are not for the last-iterate, but for the averaged policies. I was pointing to the fact that this is not what Theorem 1 in their submission claims -- theorem 1 claims convergence for the last iterate, which is not what is shown in the proof in the supplement, where the overlines denoting the average are dropped from the notation "to alleviate the presentation". Re: point(1): I continue to maintain the results are incremental over the results in [68] (in the paper) -- the consideration of common noise does not pose a significant challenge in my opinion as the analysis is for the mean field game (and hence for the aggregate distribution) and the convergence is over the fictitious play time t, and not over the game time n. For instance, in the proof in the supplement, the terms for different game times n never interact -- so in essence the results only require the Fokker-Planck equations for the aggregate distribution. Re: point (2) -- based on the authors response and other reviewers comments, I am willing to accept that this might be milder than other assumptions in the literature.

Strengths: 1. The authors allow for common noise to affect the state transitions of all agents in their model. 2. Use of an exploitability metric to capture the convergence of learning algorithm

Weaknesses: 1. The results seem incremental, and lack substantial novelty. 2. The monotonicity conditions to achieve O(1/t) convergence rate also imply uniqueness of equilibria, and thus seem to be quite strong. 3. The convergence is only established for the averaged policies, and not for the last-iterate.

Correctness: Yes.

Clarity: The notation for the averaged policies (\bar{\pi})$ is replaced by the non-averaged $\pi$ in the statement of Theorem 1, but this change is only made explicit in the proof in Appendix A (there is a passing reference right before the theorem statement). The paper would improve if this fact is made transparent.

Relation to Prior Work: Yes.

Reproducibility: Yes

Additional Feedback:

[Author Response · NeurIPS 2020]

**R#1:** Thanks for stressing the strengths of the paper (a complete theory of FP in MFG and a rich empirical evaluation).We first address the stated weaknesses. **W1: Short presentation of FP**. We'll improve it for the final version by adding formal def of the best response and explaining why an arbitrary policy between $[0,1]$ is needed for init purpose. **W2: Gap between CTFP and practical algs**: We'll add the following discussion to the paper. We chose to provide an analysis in continuous time because it provides convenient mathematical tools allowing to exhibit state of the art convergence rate. The convergence rate in discrete time is still an open problem even for 2-players games, but would be an interesting research question (there is a known conjecture in $O(1/\sqrt{t})$ [75]). **Detailed comments**: **(1)** We acknowledge that some useful details should be moved from appx to the main text for the sake of clarity. E.g. the computation of the Best Response (BR) and the population distribution (*cf.* Appx) are both used in FP (Alg. 1), which is implemented in two different settings: a model-based and a model-free approach. The model-based uses Backward Induction (BI, Alg. 4) and an exact calculation of the population distribution (Alg. 5). The model-free approach uses $Q$-learning (Alg. 2) and a sampling-based estimate of the distribution (Alg. 3). As suggested, we will add the update rules of the $Q$-function of both methods in the main text. We will clarify how the distribution $\hat{\mu}_n^\pi$ (Alg. 3) is used in Alg. 2 by using proper notations. $Q$-learning and BI approximate the BR against $\bar{\mu}^j$ (mean distribution), which needs to be clarified: we will add a line in Alg. 1 $\bar{\mu}^j = \frac{j-1}{j}\bar{\mu}^{j-1} + \frac{1}{j}\hat{\mu}^j$ (so here, $\hat{\mu}_n^\pi$ and $\hat{\mu}^j$ are the same). In Alg. 2, the $\mu$ of the input can be any distribution ($\mu = (\mu_k)_k$) but we use the mean population distribution $\bar{\mu}^j$ (from the previous FP step) in our setting. **(2) Randomization:** As we use $\bar{\mu}^j$ we don't need to select the policy uniformly over previously obtained policies. Also, we already do employ randomized strategies (for the model-free), with $\varepsilon$-greedy exploration parameter set to 0.2 (l.140). Authors of [64] use a softmax to ensure the regularity needed in their proof. To the best of our understanding, Angiuli *et al.* use $\varepsilon$-greedy action because the updates of $Q$ and $\mu$ are intertwined, so the exploration/exploitation are mixed. In Alg.2, the $Q$-learning (with exploration) and the action to update the distribution (with pure exploitation) are separated. Furthermore, the stochasticity of the environment (noise $\epsilon_n$) adds randomization. Note that randomization is not necessary in model-based as the BR and population distribution are computed exactly (which also bridges the gap between model-based and the theory). Adding $\varepsilon$-randomization or a softmax in the distribution update is an interesting direction. **Exploitability:** Please notice that, because it scales with rewards, its absolute value is not meaningful. This quantity is game dependent and hard to re-scale without introducing other issues (dependence on the initial policy if we re-normalize with the initial exploitability for example). But it decreases by a large factor compared with the initial value. **(3)** The problem of error propagation is addressed in [51] (see Eq. 7). However, [51] does not provide any rate for discrete time FP. As opposed to this work, we focused in getting a convergence rate for CTFP without approximations (in a wider set of settings than in [51]). Surprisingly, these rates do not seem to be too off in practice. We also introduce a new theory of common noise for the two practical algs (*c.f.* R#3). **(4)** We will improve on that transition stating that to go from continuous to discrete time we simply replace sums by integral and difference equations by differential equation (inclusion to be precised). The "watershed" region is necessary to make sure the differential equation is defined on a closed set (here $[1,+\infty[$). Without it, we would only be able to define it on $]0,+\infty[$ which is not enough. **(5)** We apologize for the too short Sec.3. We'll rewrite it with elements from appx A. Even if not directly used, we felt that the equation involving $\pi_n^t$ was important as it is easier to manipulate policies compared to distribution over states. **(6)** Our common noise can be history dependent (i.e., no assumption on it). In the experiment of Sec.6, the common noise is stationary and i.i.d. Common noises affect the transition probability of the distribution, which is then *random* (it is not the case with only idiosyncratic noise).

**R#3:** We are grateful for the positive comments acknowledging the importance of common noise in MFGs and MARL, and on the fact that our contribution bridges the gap between MFG and tools from algorithmic game theory such as exploitability. **W1: connections with MARL examples:** Actually our numerical examples are strongly motivated by classical examples in the RL literature. For instance, the beach bar process example is a simplified version of the well known Santa Fe bar problem, which has received a strong interest in the MARL community, see e.g. [Farago et al, Fair and Efficient Solutions to the Santa Fe Bar Problem (2002)]. Similarly, the maze is motivated by swarm motion models from the distributed robotics MARL literature. We will stress this point and add references in the revision. **Other works:** Thank you for pointing out these relevant references, that we will cite as well. Note however that, compared with these works, our paper provides a rigorous rate of convergence, and covers the common noise setting. Last but not least, our work is not limited to potential or variational MFGs as we only need the weaker monotonicity assumption.

**R#5: (1)** We strongly disagree about the lack of novelty and incremental nature of our work, and would have appreciated some argument for this harsh comment. We would like to stress that the other two referees have acknowledged the novelty of work (rate of convergence, common noise, etc.). **(2)** The monotonicity assumption is classical in the MFG literature and much weaker than assumptions made in other works (regularity and smallness of the coefficients in [64], potential structure in [84], etc.). Also, R#1 considers these assumptions as mild. **(3)** This is the very principle of the fictitious play to obtain convergence for averaged policies. We would appreciate any reference where it is not the case.

[Meta-Review · NeurIPS 2020]

The reviewers acknowledge that the paper established convergence of continuous-time fictitious play (CTFP) in mean-field games with mild assumptions. The numerical results are also interesting. The authors should consider revising the paper according to the revision suggestions.